# Study on Residual Vibration Suppress of a 3-DOF Flexible Parallel Robot Mechanism

**DOI:** 10.3390/s18124145

**Published:** 2018-11-26

**Authors:** Qinghua Zhang, Qinghua Lu, Xianmin Zhang, Junjun Wu

**Affiliations:** 1Department of Mechatronics, Foshan University, Foshan 528000, China; zhangqh@fosu.edu.cn (Q.Z.); jjunwu@fosu.edu.cn (J.W.); 2Key Laboratory of Precision Equipment and Manufacturing Technology of Guangdong Province, South China University of Technology, Guangzhou 510641, China; zhangxm@scut.edu.cn

**Keywords:** residual vibration control, flexible parallel robot mechanism, strain and strain rate feedback, experimental study

## Abstract

Residual vibration suppression of a 3-DOF flexible parallel robot mechanism is implemented in this paper. Considering the direct and inverse piezoelectric effect of PZT (lead zirconium titanate) material, a general motion equation is established which includes an input equation of PZT actuators and an output equation of PZT sensors. A strain and strain rate feedback (SSRF) controller is designed based on the established general motion equation. A numerical simulation is implemented to verify the effectiveness of the SSRF controller in driving the proposed robotic mechanism. The simulation results reveal that the SSRF controller can decrease the elastic vibration displacement of the flexible links rapidly and improve the position accuracy of the moving platform. In the experimental study, one scheme with three passive flexible links is controlled by the SSRF controller at the same time as the performance of the introduced solutions. The experimental results show that the strain and strain rate feedback controller is able to effectively suppress the residual vibration of the 3-DOF flexible parallel robot mechanism. The results of the numerical simulation and experiment are completely consistent.

## 1. Introduction

Saving energy and improving industrial productivity is a consistent target around the world. During the past few decades, flexible parallel robots have been intensively studied because of their excellent performances which have the advantages of light weight, low energy consumption, dexterous operation, and so on. However, the elastic vibration issue of such robots is serious due to their flexible components, as well as their inertial and joint driving forces. Therefore, establishing nonlinearity, rigid-elastic coupling multibody system dynamics and suppressing the unwanted elastic vibration is currently a very significant and challenging problem to solve in this research field.

Dynamic modeling and active vibration control of flexible manipulators have been paid considerable attention, as indicated in a number of survey papers [1,2,3,4,5,6,7]. Although there are many theoretical and experimental studies of active vibration control in flexible structures and simple flexible manipulators [7,8,9], the active vibration control of flexible parallel robot mechanisms is less studied due to its complexity.

PZT materials are able to both dampen vibration and measure the vibration of distributed parameter systems because of their direct and inverse piezoelectric effect [10]. The PZT sensor has the advantages of fast response, good linearity, low energy consumption, low cost, and easy manufacturing. Therefore, they have been widely used for active vibration control of flexible structures [11,12,13,14,15].

Qiu et al. [16,17] studied the active vibration control of a simple flexible plate using a non-collocated acceleration sensor, piezoelectric patch actuator, and sensor, obtaining an excellent control effect. Lin et al. [18] employed piezoelectric sensors and actuators for suppressing vibrations of thin flexible plate held vertically. In their approach, an adaptive fuzzy sliding mode control algorithm was designed and they were able to effectively suppress elastic vibrations caused by modal excitation, external disturbances, and coupling effects experimentally. Giorgio et al. [19] studied the trajectory-tracking and vibration control of highly flexible planar multi-links robot arms. A dynamic model was established based on the Hencky bar-chain model (a kind of lumped parameters technique), and the control strategy employed an optimal input pre-shaping and a feedback of the joint angles. Their approach was based on the different trajectories of the end-effecter of the robot and compared with a standard collocated Proportional-Derivative (PD) control strategy. The potential of the proposed controller was verified by numerical examples.

In comparison to theoretical study, the experimental study of the active vibration control of flexible manipulators is more challenging, especially for flexible parallel robot mechanisms. Liang et al. [20] studied a novel 2-DOF flexible parallel manipulator (PM) with multiple actuation modes, establishing the electromechanical coupling dynamic equation of the system and designing a hierarchical compound control strategy, in which a PD feedback controller was employed to achieve the trajectory tracking of the end-effector and a strain and strain rate feedback (SSRF) controller was developed to suppress the vibration of the flexible links using PZT. The numerical simulation results verified the effectiveness of the proposed control method. Unfortunately, an experimental study is not presented in this paper. Zhang et al. [21] investigated a 3-PRR flexible parallel manipulator and performed an experimental study of active vibration control of 3-PRR flexible parallel manipulators, for which the elastic vibration of flexible links during motion was suppressed by a strain rate feedback (SRF) controller and the KED assumption that the small amplitude, high-frequency structural vibrations of the manipulator have a negligible effect on its rigid body motion was utilized. Zhang et al. [22] designed a smooth adaptive sliding mode for suppressing the vibration of smart links which bounded multiple lead zirconium titanate (PZT) sensors and actuators on the flexible link. Their experimental results verified the multi-mode vibration control ability of the smooth adaptive sliding-mode control law.

We performed dynamic modeling and active vibration control of a planar 3-RRR flexible parallel robot mechanism [23,24]. The residual vibration of flexible links influences the repeat positioning accuracy of the system seriously. Thus, it is important to suppress the residual vibration of flexible links to improve the repeat positioning accuracy of the system. An experimental study of the active vibration control of planar 3-RRR parallel robot with three flexible links, each of which bonds with two pairs of PZT actuators and one PZT sensor film, is implemented in this paper. An SSRF control algorithm is adopted to suppress the elastic vibration of the flexible links of the system. Both numerical simulation and experimental studies verify that the SSRF controller can effectively suppress the elastic vibration of the flexible links. 

## 2. Dynamic Model of 3-DOF Flexible Parallel Robot Mechanism

A dynamic equation of a 3-DOF flexible parallel robot mechanism is given in this section. As seen Figure 1, first, flexible links are divided into several beam elements according to the finite element method (FEM), and then, according to the Lagrange equation, a dynamic equation of the beam element is established. Considering the constraint condition that includes rigid motion constraint, elastic deformation motion constraint, and rigid moving platform constraint, a general dynamic equation for the 3-DOF flexible parallel robot mechanism is formed by assembling the dynamic equation of the beam element.

### 2.1. Sketch of 3-DOF Flexible Parallel Robot Mechanism

Figure 2 shows a sketch of a 3-DOF flexible parallel robot mechanism that is constructed by the regular triangle moving platform $C_{1}C_{2}C_{3}$, the regular triangle static platform $A_{1}A_{2}A_{3}$, and three symmetrical kinematic chains $A_{i}B_{i}C_{i}(i=1,2,3)$. Each kinematic chain has three revolute (RRR) joints, in which “R” is an active revolute joint located at $A_{i}\left(i=1,2,3\right)$, and “R” is a passive revolute joint. $A_{1}B_{1}=A_{2}B_{2}=A_{3}B_{3},\;B_{1}C_{1}=B_{2}C_{2}=B_{3}C_{3}$, where two points O, P represent the midpoint of the regular triangles $A_{1}A_{2}A_{3}$ and $C_{1}C_{2}C_{3}$, respectively. Assuming that O−XY is the global fixed frame, $L_{1},L_{2},L_{3},L_{4}$ are the lengths of the segments $A_{i}B_{i},B_{i}C_{i},C_{i}P,\;OA_{i}\left(i=1,2,3\right)$, θ is the angle between the X axis and side $C_{1}C_{2}$ of the regular triangle $C_{1}C_{2}C_{3}$, and the parameters $\alpha_{i}$ and $\beta_{i}$ are the angles between the X axis and links $A_{i}B_{i}$, $B_{i}C_{i}(i=1,2,3)$, respectively.

### 2.2. Dynamic Modeling of the Beam Element

Figure 3 is a beam element deformation motion diagram in which O−XY is the global fixed frame and A−xy is the local moving frame with the x axis along the neutral line of the beam element and the original point *A* located at an endpoint of the beam element. $O-x^{\prime }y^{\prime }$ is an intermediate coordinate frame whose origin is same as that of O−XY and whose axes are parallel to the axes of A−xy. φ is the angle between O−XY and $O-x^{\prime }y^{\prime }$.

Assume that point C is any point on the beam element, and point $C_{0}$ is the corresponding point on the neutral line. Points $C,C_{0}$ move to $C^{\prime },C_{0}{}^{\prime }$ due to flexible link elastic deformation motion, respectively, setting $e_{f}={\left[e_{1},e_{2},e_{3},e_{4},e_{5},e_{6},e_{7},e_{8}\right]}^{T}$, which is the generalized elastic coordinate vector of two endpoints of the beam element in the A−xy frame. Here, $e_{1}$ and $e_{5}$ are the axial displacements of two endpoints A and B, respectively; $e_{2}$ and $e_{6}$ are the transverse displacements; $e_{3}$ and $e_{7}$ are elastic rotational angles; $e_{4}$ and $e_{8}$ are section curvatures. Then, the elastic deformation displacement of point C in A−xy is given by Reference [23].
(1)$$v\left(x,y,t\right)=\left[\begin{array}{c} v_{10}\left(x,t\right)-y\frac{\partial v_{20}(x,t)}{\partial x}\\ v_{20}\left(x,t\right) \end{array}\right]=N\left(x,y\right)e_{f}$$
where $v_{10}\left(x,t\right),v_{20}\left(x,t\right)$ are axial deformation displacement and lateral deformation displacement of point $C_{0}$, respectively, the coefficient of $v_{10}\left(x,t\right)$ is linear function about x, the coefficient of $v_{20}\left(x,t\right)$ is five order hermit interpolation function about x.

Then the displacement of $C^{\prime }$ can be expressed in O−XY by
(2)$$r_{c^{\prime }}=r_{A}+R(\varphi )\left(r_{0}+N(x,y)e_{f}\right)$$
where $r_{A}$ is the coordinate of the point A in O−XY, $R(\varphi )=\left[\begin{array}{cc} \cos\varphi & -\sin\varphi \\ \sin\varphi & \cos\varphi \end{array}\right]$ is the direction cosine matrix, The vector $r_{0}={\left[\begin{array}{cc} x & y \end{array}\right]}^{T}$ is the location coordinates of the point C in the A−xy system, ‘T’ indicates the matrix transpose, 

Taking the first derivative of Equation (2), yield:(3)$${\dot{r}}_{C^{\prime }}={\dot{r}}_{A}+R_{\varphi }\left(r_{0}+Ne_{f}\right)\dot{\varphi }+R(\varphi )N(x,y){\dot{e}}_{f}=S\dot{q}$$
where $R_{\varphi }=\frac{\partial R(\varphi )}{\partial \varphi }$, $S=\left[\begin{array}{ccc} I & R_{\varphi }\left(r_{0}+N(x,y)e_{f}\right) & R(\varphi )N(x,y) \end{array}\right]$, $q={\left[\begin{array}{ccc} r_{{}_{A}}^{T} & \varphi & e_{f}^{T} \end{array}\right]}^{T}$.

So, kinetic energy of the beam element can be expressed by
(4)$$\begin{array}{ll} T & =\frac{1}{2}\int_{V}\rho {\dot{r}}_{C^{\prime }}^{T}{\dot{r}}_{C^{\prime }}dV+\int_{0}^{L_{e}}\frac{1}{2}{\dot{\varphi }}^{2}\left(x,t\right)dJ_{c}+\frac{1}{2}(m_{A}{\dot{r}}_{A^{\prime }}^{T}{\dot{r}}_{A^{\prime }}+m_{B}{\dot{r}}_{B^{\prime }}^{T}{\dot{r}}_{B^{\prime }})+\frac{1}{2}\left(J_{A}{\dot{\varphi }}^{2}\left(0,t\right)+J_{B}{\dot{\varphi }}^{2}\left(L,t\right)\right)\\ & =\frac{1}{2}{\dot{q}}^{T}m\dot{q} \end{array}$$
where $m_{A},J_{A},m_{B},J_{B}$ are lumped mass and lumped moment of inertia of two endpoints A,B of beam element.

According to Material mechanics knowledge, neglecting shear strain energy and buckling strain energy, strain energy of the beam element can be expressed
(5)$$\begin{array}{l} U=\frac{1}{2}\int_{0}^{l}EI(x){\left(v_{20}^{\prime \prime }(x,t)\right)}^{2}dx+\frac{1}{2}\int_{0}^{l}ES(x){\left(v_{10}^{\prime }(x,t)\right)}^{2}dx\\ \begin{array}{ccc} & =\frac{1}{2}q^{T}kq & \end{array} \end{array}$$
where $v_{20}^{\prime \prime }(x,t)=\frac{\partial^{2}v_{20}(x,t)}{\partial x^{2}}$, $v_{10}^{\prime }(x,t)=\frac{\partial v_{10}(x,t)}{\partial x}$, *E* is elastic modulus of materials, I(x), S(x) are cross-sectional moment of inertia and cross-sectional area of the beam element, respectively.

According to Lagrange’s equation, dynamic equation of the beam element can be written as
(6)$$m\ddot{q}+kq=p_{e}+p_{v}$$
where $p_{e}$ are the generalized external forces and $p_{v}$ is the quadratic velocity vector that contains the gyroscopic and the coriolis force components, respectively.

Define the coordinate transformation matrix ***B***, then
(7)$$e_{f}=Bu$$
where u is the elemental nodal coordinate vector in the *O*-*XY* frame. Taking the first and the second derivative on (8) with respect to t, yields
(8)$$\begin{array}{l} {\dot{e}}_{f}=B_{\varphi }u\dot{\varphi }+B\dot{u}\\ {\ddot{e}}_{f}=B_{\varphi \varphi }u{\dot{\varphi }}^{2}+2B_{\varphi }\dot{u}\dot{\varphi }+B_{\varphi }u\ddot{\varphi }+B\ddot{u} \end{array}$$

Substitute (7) and (8) into (6) and premultiply by matrix $\left[\begin{array}{ccc} I & 0 & 0\\ 0 & 1 & u^{T}B_{\varphi }^{T}\\ 0 & 0 & B^{T} \end{array}\right]$, then
(9)$$\bar{m}(q)\ddot{q}+\bar{k}q=\bar{p}(q,\dot{q})$$
where $q=\left[\begin{array}{ccc} r_{A}^{T} & \varphi & u^{T} \end{array}\right]$.

According to generalized coordinate variables, Equation (10) can be divided to rigid subsystem and elastic subsystem, namely
(10)$$m_{r}\ddot{\varphi }=p_{r}\left(q,\dot{q}\right)+\tau_{r}$$
(11)$${\tilde{m}}_{f}\ddot{u}+c_{f}\dot{u}+k_{f}u=p_{f}$$

Generalized coordinates vector q are formed by rigid-body motion coordinates and elastic coordinates. In generally, the vector q is not independent, existing some constraints relations that include the rigid body motion constraint, the elastic deformation motion constraint and dynamic constraints of the moving platform. As shown in Figure 2, rigid body motion constraint can be expressed by
(12)$$OA_{i}+A_{i}B_{i}+B_{i}C_{i}+C_{i}P+PO=0$$

The moving platform and flexible links $B_{i}C_{i}$ are joined together by the rotate joint $C_{i}$, so, existing constraint relation between elastic deformation displacements of point $C_{i}$ and elastic deformation displacements of the moving platform, namely
(13)$$\left(\begin{array}{c} \Delta X_{C_{i}}\\ \Delta Y_{C_{i}} \end{array}\right)=\left(\begin{array}{c} X_{C_{i}^{\prime }}\\ Y_{C_{i}^{\prime }} \end{array}\right)-\left(\begin{array}{c} X_{C_{i}}\\ Y_{C_{i}} \end{array}\right)=\left[\begin{array}{cc} I & T_{1}{\left(\begin{array}{c} -y_{C_{i}}\\ x_{C_{i}} \end{array}\right)}_{P} \end{array}\right]\left(\begin{array}{c} \Delta X_{P}\\ \Delta Y_{P}\\ \varepsilon \end{array}\right)$$
(14)$$\varepsilon =\sum_{i=1}^{3}\Delta \eta_{i}$$
where $\Delta X_{C_{i}},\Delta Y_{C_{i}},\Delta \eta_{i}$ are elastic deformation displacements and angle of the point $C_{i}$, $\Delta X_{P},\Delta Y_{P},\varepsilon$ elastic deformation displacements and angle of are moving platform, I is 2×2 unit matrix, $T_{1}=\left[\begin{array}{cc} \cos\varphi & -\sin\varphi \\ \sin\varphi & \cos\varphi \end{array}\right]$.

Assuming that $F_{i}$ is the generalized joint constraint anti-force that exert to the moving platform. Then, dynamic constraint of the moving platform can be expressed by
(15)$$\left[\begin{array}{ccc} M_{v} & & \\ & M_{v} & \\ & & J_{v} \end{array}\right]\left[\begin{array}{c} \Delta {\ddot{X}}_{P}\\ \Delta {\ddot{Y}}_{P}\\ \ddot{\varepsilon } \end{array}\right]=\sum_{i=1}^{3}F_{i}-\left[\begin{array}{c} M_{v}{\ddot{X}}_{P}\\ M_{v}{\ddot{Y}}_{P}\\ J_{v}\ddot{\phi } \end{array}\right]$$
where $M_{v},J_{v}$ are the mass and moment of inertia of the moving platform.

Now, the elastic dynamic equation that describes the dynamic characteristic of system can be formed through assembling element dynamic Equation (11) and considering constraint equations.
(16)$$M\left(\alpha \right)\ddot{U}+C\left(\alpha ,\dot{\alpha }\right)\dot{U}+K\left(\alpha ,\dot{\alpha },\ddot{\alpha }\right)U=Q\left(\alpha ,\dot{\alpha },\ddot{\alpha }\right)$$
where $U={\left[U_{11},U_{12},\ldots ,U_{1n},\ldots ,U_{3n},\Delta X_{P},\Delta Y_{P},\Delta \varepsilon \right]}^{T}$ is generalized coordinate vector of the system, $\alpha ={\left[\alpha_{1},\alpha_{2},\alpha_{3}\right]}^{T}$ is nominal motion variable.

## 3. Strain and Strain Rate Feedback Control Algorithm

Due to the direct and inverse piezoelectric effect of PZT material, PZT material is widely used for designing active vibration controllers of flexible multibody systems. Assuming that a PZT sensor film is perfectly bonded on the flexible beam, as shown in Figure 4, the direct piezoelectric equation of a PZT sensor can be expressed as [25]:(17)$$V_{se}=k_{se}\left(\frac{\partial v_{{}_{2}}^{2}(x_{k},t)}{\partial x^{2}}\right)=k_{se}N_{2}{{}^{\prime \prime }}^{T}(x_{k})e_{f}$$
where $k_{se}=\frac{E_{s}d_{31}w_{s}}{C_{s}}\left(L_{s}+L_{e}\right)$, $E_{s}$ is the Young’s modulus of the piezoelectric sensor, $d_{31}$ is the piezoelectric constant, $w_{s}$ is the width of the piezoelectric sensor, $L_{s}$ is the length of the piezoelectric sensor, $C_{s}$ is the capacitance of the piezoelectric sensor, $L_{e}$ is the length of the beam element, $x_{k}$ is the x direct coordinates of the midpoint of the kth sensor in the element coordinate frame, $N_{2}\left(x\right)$ is the column vector of the second line of N(x), ″ denotes the second derivative to the variable x, and ‘T’ expresses the inversion of the matrix.

As shown in Figure 5, two of the same PZT actuator films are perfectly bonded onto the upper and lower surfaces of the beam element in the same position and the polarization direction of the two PZT actuators is also the same. Two adhesive surfaces are electrically grounded, and the other two surfaces are linked to an input voltage by wire. According to the inverse piezoelectric effect of the PZT material, the torque of the two ends of the PZT actuator pair is given by reference [26]
(18)$$M_{ae}(x_{k_{1}})=-M_{ae}(x_{k_{2}})=-k_{ae}V_{ae}$$
where $k_{ae}=d_{31}E_{p}w_{p}(t_{p}+t_{b})$, $V_{ae}=V_{input}$, $x_{k_{1}}$ is the x direct coordinate of the left end of the kth PZT actuator, and $x_{k_{2}}$ is the *x* direct coordinate of the right end of the kth PZT actuator. $E_{p}$ is the Young’s modulus of the PZT actuator, $w_{p}$ is the width of the PZT actuator, $t_{p}$ is the thickness of the PZT actuator, and $t_{b}$ is the thickness of the beam element.

Assuming that there are m PZT actuator pairs and p PZT sensor films in this system, integrating all of the inverse piezoelectric equations of the PZT actuator pairs and Equation (16) yields:(19)$$M\ddot{U}+C\dot{U}+KU=Q+\bar{K}V_{in}$$
where $\bar{K}$ is the input matrix and $V_{in}$ is the m dimension input vector.

Assembling all of the direct piezoelectric equations of the PZT sensor films yields:(20)$$V_{s}=C_{s}U$$
where $V_{s}$ is the *p* dimension output vector, and $C_{s}$ is the output matrix. According to Equations (19) and (20), the general motion equation of the system can be expressed as:(21)$$\{\begin{array}{c} M\ddot{U}+C\dot{U}+KU=Q+\bar{K}V_{in}\\ V_{s}=C_{s}U \end{array}$$

In this paper, the goal is to suppress the residual vibration of flexible links of a parallel mechanism. Therefore, the mass matrix M is a positive definite symmetric constant matrix, the stiffness matrix K is a symmetric constant matrix, and Q is a zero vector, assuming that the damping matrix C barely includes structural damping. Considering that the residual vibration of the system is decided by its lower-order modes, the real modal method is utilized for extracted lower-order modes of the controlled system. Assuming that the first *r* lower-order modes are suppressed, the general motion equation of the system can be simplified as:(22)$$\{\begin{array}{c} {\ddot{\eta }}_{r}+diag(2\zeta_{1}w_{1},2\zeta_{1}w_{1},\ldots ,2\zeta_{r}w_{r}){\dot{\eta }}_{r}+diag(w_{1}^{2},w_{2}^{2},\ldots ,w_{r}^{2})\eta_{r}=K_{r}V_{input}\\ V_{s}=C_{r}\eta_{r} \end{array}$$
where $\eta_{r}$ is the first *r* lower-order mode coordinate.

A strain and strain rate feedback controller is adopted to suppress the residual vibration of the flexible links. Here, the output voltage of the sensors and its derivation are fed back into the corresponding actuator pairs, yielding:(23)$$V_{input}=-K_{d}{\dot{V}}_{s}-K_{p}V_{s}=-K_{d}C_{r}{\dot{\eta }}_{r}-K_{p}C_{r}\eta_{r}=-{\bar{K}}_{d}{\dot{\eta }}_{r}-{\bar{K}}_{p}\eta_{r}$$
where $K_{d},K_{p}$ represent the feedback gain matrix.

Substituting Equation (23) into (22), the motion equation of the controlled system can be expressed as:(24)$${\ddot{\eta }}_{r}+\left(diag(2\zeta_{1}w_{1},2\zeta_{1}w_{1},\ldots ,2\zeta_{r}w_{r})+K_{r}{\bar{K}}_{d}\right){\dot{\eta }}_{r}+\left(diag(w_{1}^{2},w_{2}^{2},\ldots ,w_{r}^{2})+K_{r}{\bar{K}}_{p}\right)\eta_{r}=0$$

Assuming that the damping ratio coefficient $\zeta_{i}$ is accelerated to ${\bar{\zeta }}_{i}$ and the natural frequency $w_{i}$ is accelerated to ${\bar{w}}_{i}$, then the feedback gain matrix $K_{d},K_{p}$ can be expressed as:(25)$$K_{d}=K_{r}^{+}diag\left(2\left({\bar{\zeta }}_{1}-\zeta_{1}\right)w_{1},2\left({\bar{\zeta }}_{2}-\zeta_{2}\right)w_{2},\ldots ,2\left({\bar{\zeta }}_{r}-\zeta_{r}\right)w_{r}\right)C_{r}^{+}$$
(26)$$K_{p}=K_{r}^{+}diag\left({\bar{w}}_{1}^{2}-w_{1}^{2},{\bar{w}}_{2}^{2}-w_{2}^{2},\ldots ,{\bar{w}}_{r}^{2}-w_{r}^{2}\right)C_{r}^{+}$$
where $K_{r}^{+}$ and $C_{r}^{+}$ are the generalized inverse matrices of $K_{r}$ and $C_{r}$, respectively.

According to Equation (24), the strain and strain rate feedback control leads to the increase of the damping and stiffness of the system.

## 4. Numerical Simulation Analysis

In this section, a numerical simulation analysis of the planar 3-DOF flexible parallel manipulator is introduced. Assuming that all three passive links are flexible and other components that include three active links, joints, and the moving platform are rigid, each flexible link is divided into three equal beam elements. Moreover, each passive flexible link bonds two pairs, with each pair consisting of a PZT actuator and a PZT sensor film. Meanwhile, a PZT sensor film is centered and bonded onto the second beam element of the passive flexible links and two pairs of PZT actuators are symmetrically pasted in the first and third beam elements of the passive flexible links. Thus, there are three PZT sensors and six pairs of PZT actuators in the whole system. The material of the 3-RRR system is aluminum alloy. The thickness of the moving platform $c=0.0034\;m,\;L_{1}=0.245\;m,\;L_{2}=0.242\;m,\;L_{3}=0.108\;m,\;L_{4}=0.4\;m$, the lumped mass of the joint $M_{v}=0.02\;\mathrm{kg}$, and the lumped moment of inertia $J_{v}=0.00005\;\mathrm{kg}\;m^{2}$. The other parameters of flexible links are defined in Table 1. 

Assuming that the motion trajectory of the moving platform of the system is granted by Equation (27), we obtain:(27)$$\{\begin{array}{c} X_{P}=0.05\sin(20\pi t/3)\\ Y_{P}=0.05\cos(20\pi t/3)\\ \theta =0 \end{array}$$

In this paper, we only focus on the residual vibration that is the vibration of flexible links after the nominal motion of the system has stopped, which is caused by the impact of the system and the coupling of the motor.

Figure 6 shows the maximum elastic displacements of three active links and three passive links along the *X* direction. Figure 7 shows the maximum elastic displacements of three active links and three passive links along the *Y* direction. Lastly, Figure 8 shows the maximum elastic rotation angle of three active links and three passive links. It can be seen that the elastic displacements and rotation angles of three passive links are obviously greater than those of the corresponding three active links in all these figures. Therefore, the above assumption that the three active links are rigid and the three passive links are flexible is verified.

Figure 9 depicts the elastic displacements of the moving platform along the *X* direction. From Figure 9, it can be seen that the vibration displacements along the X direction quickly reduce to around 0 when the SSRF controller is adopted. Figure 10 shows the elastic displacements of the moving platform along the Y direction; the same conclusion that the vibration displacements along the Y direction quickly reduce to around 0 after exerting control can be obtained. Figure 11 shows the elastic rotation angles of the moving platform in the XY plane, and the same conclusion is also obtained. Detailed information is given in Table 2, concerning Figure 9, Figure 10 and Figure 11. Table 2 reflects the time-consuming data concerning the amplitude of elastic displacements and elastic rotation angles, and can more clearly illustrate the effectiveness of the SSRF controller.

Figure 12a–f show the control voltages exerted on PZT actuator pairs 1, 2, 3, 4, 5, and 6, respectively. The control voltage of all PZT actuator pairs is limited to the range of −250 V to 250 V. The changes in the control voltage are that same as those of the vibration displacement; larger control voltages are required when the vibration displacements are large, while when the vibration displacements decrease, the required control voltages also decrease.

## 5. Vibration Experiment Study

Figure 13 is the flow diagram of the planar 3-RRR flexible parallel robot system testing experiment. Firstly, PZT sensors collect the vibration signals of the flexible links of the 3-RRR flexible parallel robot system, and the vibration signals are amplified through a charge amplifier. Then, these signals are sent to the SSRF controller through ADC ports of dSPACE, so the SSRF controller can calculate the control output. The control voltage is the exerted on PZT actuator pairs that are bound to the flexible links of the planar 3-RRR flexible parallel robot system through the PZT drive power. Finally, the elastic vibration of the flexible links is suppressed, which will eventually improve the positioning accuracy of the moving platform of the system.

### 5.1. Experiment Platform Introduction

The experimental setup of the active vibration control system, as shown in Figure 14, consists of a 3-DOF flexible parallel manipulator, a static platform, three Yaskawa servo motors, and SHIMPO reducers (with reduction ratio of 1:5). The three passive links are all flexible, with identical dimensions of 252 mm × 25 mm × 3 mm. The three active links are all rigid, with identical dimensions of 254 mm × 25 mm × 10 mm. The links are insulated by surface oxidation. The static platform of the 3-DOF flexible parallel manipulator consists of a marble pedestal, a steel-framed structure, and a rectangular steel plate. The experimental setup also includes PZT actuators and PZT sensors, a PZT drive power, a charge amplifier, a real-time semi-physical simulation system—dSPACE, and an industrial computer. The dimensions of each PZT actuator are 50 mm × 25 mm × 2 mm, and the dimensions of each PZT sensor are 30 mm × 15 mm × 1 mm. The piezoelectric constant of the piezoelectric actuator/sensor is *d*_31_ = 1.85 × 10^−10^ c/N, and the Young’s modulus of the piezoelectric actuator/sensor is *E*_s_ = 1.17106 MP. The smart beam in Figure 15 has two pairs of PZT actuators symmetrically bonded on the surface of the flexible beam and one PZT sensor film bonded on its middle.

### 5.2. Experimental Result and Analysis

The motion trajectory of the moving platform of the system is designed as follows:(28)$$\{\begin{array}{c} X_{P}=0\\ Y_{P}=0.05t-\frac{1}{40\pi }\sin(2\pi t)\\ \theta =0 \end{array}$$
(29)$$\{\begin{array}{c} X_{P}=0\\ Y_{P}=0.05(1-t)-\frac{1}{40\pi }\sin(2\pi t)\\ \theta =0 \end{array}$$
(30)$$\{\begin{array}{c} X_{P}=0.05\sin(20\pi t/3)\\ Y_{P}=0.05\cos(20\pi t/3)\\ \theta =0 \end{array}$$

*First step:* The midpoint of the moving platform moves to the coordinate (0, 0.05) from the origin of the fixed coordinate system along the linear trajectory of Equation (28); the process takes 1 s.

*Second step:* It then moves 0.3 s along the circular trajectory of Equation (29), and stops for 2.7 s.

*Third step:* The moving platform returns to the initial position along the linear trajectory of Equation (30). The entire motion period takes 5 s.

The PZT sensor films are bonded onto passive flexible links, as shown in Figure 14. The vibration signal of one PZT sensor film provides feedback to two pairs of PZT actuators. Thus, there are three PZT sensors and six pairs of PZT actuators in the whole system. Because we studied the residual vibration of the flexible links in this paper, the 2.7-s stopping period was chosen as the data collection time, and the sampling frequency was set at 10,000 Hz in the MATLAB/Simulation. Thus, each sensor collects 27,000 data points.

Figure 16 shows the vibration signals of passive flexible link 1 for the uncontrolled and SSRF controlled systems. Figure 17 shows the vibration signals of passive flexible link 2 for the uncontrolled and SSRF controlled systems. Figure 18 shows the vibration signals of passive flexible link 3 for the uncontrolled and SSRF controlled systems. As shown in Figure 16, Figure 17 and Figure 18, the elastic vibrations of the passive flexible links 1, 2, and 3 are effectively suppressed through SSRF control. Because coupling of the elastic vibration between the three passive flexible links exists, the vibration amplitude of each flexible link and the control effect on each link are slightly different.

Figure 19 shows the control voltages exerted on pairs of PZT actuators 1, 2, 3, 4, 5, and 6. From Figure 19, it can be seen that the control voltages have the same variation as the vibration displacement of the same passive flexible link. Because the output voltage range of the PZT drive power is limited at ±150 V, when the feedback voltages of DAC of dSPACE go beyond ±10 V, the output voltages of the PZT drive power will be intercepted to ±150 V (feedback voltages are amplified 15 times). Comparing the six figures of Figure 19, it can also be seen that the control voltages of PZT actuators 1, 2 are much higher than those of the other PZT actuators, because the vibration amplitude of flexible link 1 is larger than those of flexible links 2, 3, thus much higher control voltages are required for flexible link 1.

## 6. Conclusions

An active vibration control algorithm was studied in this paper. The strain and strain rate feedback controller were designed based on the general motion equation, which considers the direct and inverse piezoelectric equations of the PZT actuator and the PZT sensor. Theoretical and experimental studies were implemented on a planar 3-RRR flexible parallel robot mechanism. In the study, three passive flexible links were assumed to be controlled at the same time. Both the numerical and experimental results indicate that the proposed controller can effectively suppress the elastic vibration of the flexible links, and the control performance of the SSRF can be improved through adjusting the parameters $K_{d},K_{p}$. The experimental study further verified the effectiveness of the SSRF algorithm, but the theorical model was simplified by the scientific and reasonable method employed. Thus, some factors were ignored because of the complexity of parallel mechanisms, which include the electromechanical coupling of the servo motor and parallel mechanism, the torsional deformation of the flexible links, friction, time-lag, gaps between joints, and so on. Therefore, the experimental results did not match the simulation results of the theoretical model completely. These results can provide guidelines for later research work on the elastic vibration control of distributed flexible elements of multibody systems.

## Figures and Tables

**Figure 1 sensors-18-04145-f001:**
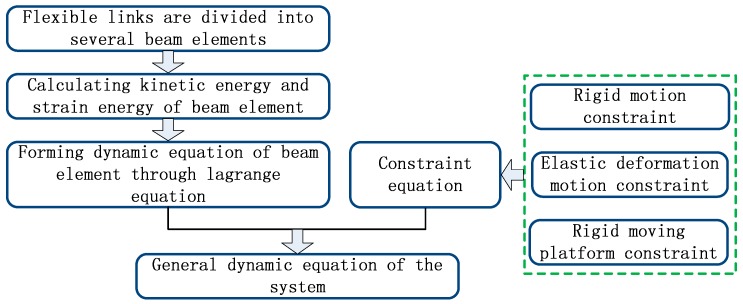
Modeling process of the 3-DOF flexible parallel robot mechanism.

**Figure 2 sensors-18-04145-f002:**
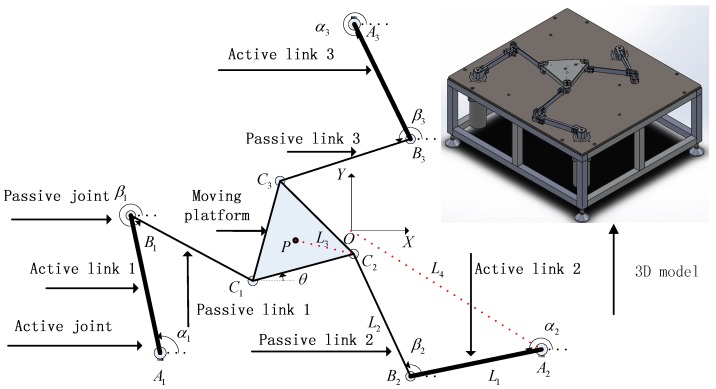
Sketch of the 3-DOF flexible parallel robot mechanism.

**Figure 3 sensors-18-04145-f003:**
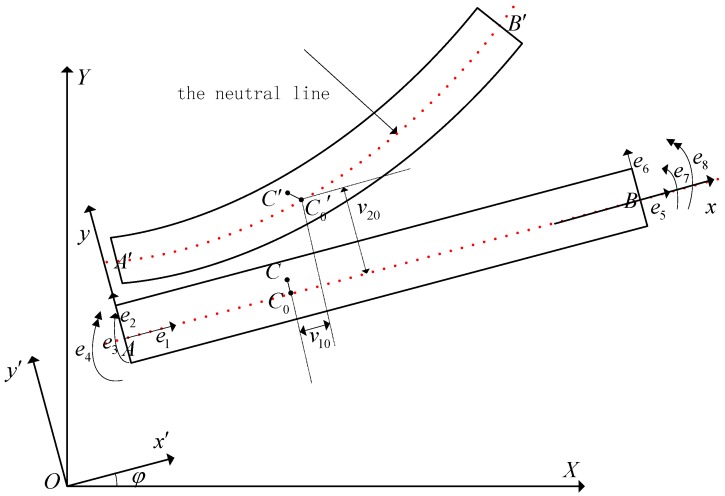
Beam element.

**Figure 4 sensors-18-04145-f004:**
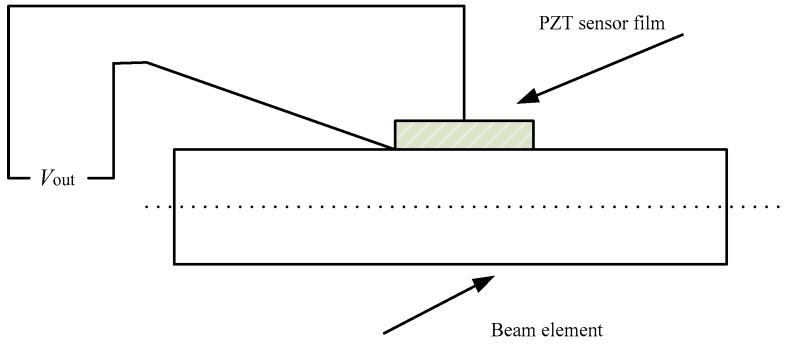
One PZT sensor film is bonded on the beam element.

**Figure 5 sensors-18-04145-f005:**
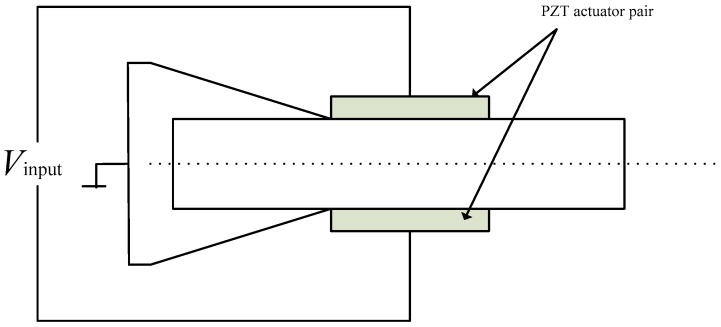
One PZT actuator pair is bonded on the beam element.

**Figure 6 sensors-18-04145-f006:**
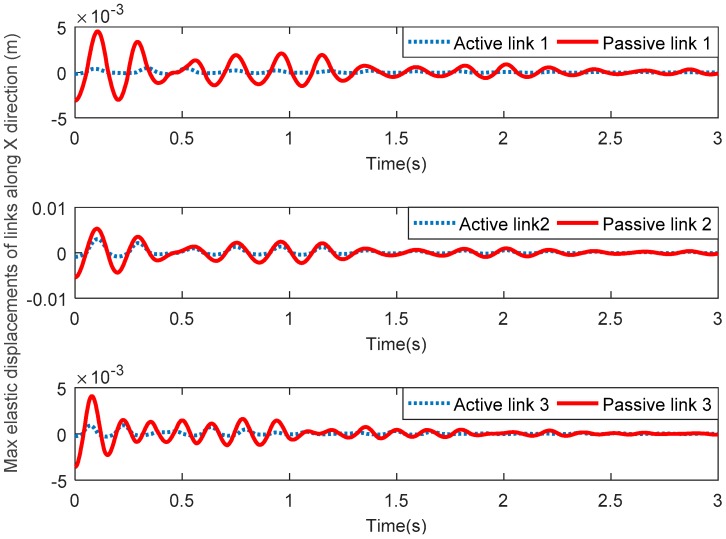
Maximum elastic displacements of active links and passive links along the *X* direction.

**Figure 7 sensors-18-04145-f007:**
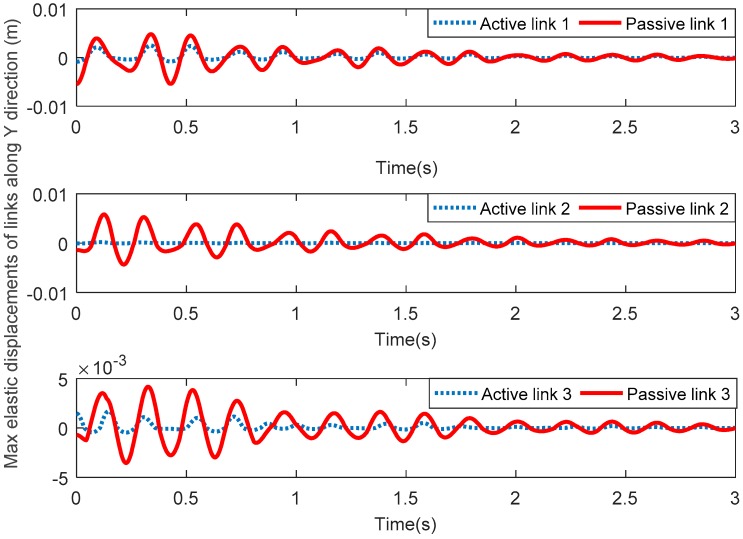
Maximum elastic displacements of active links and passive links along the *Y* direction.

**Figure 8 sensors-18-04145-f008:**
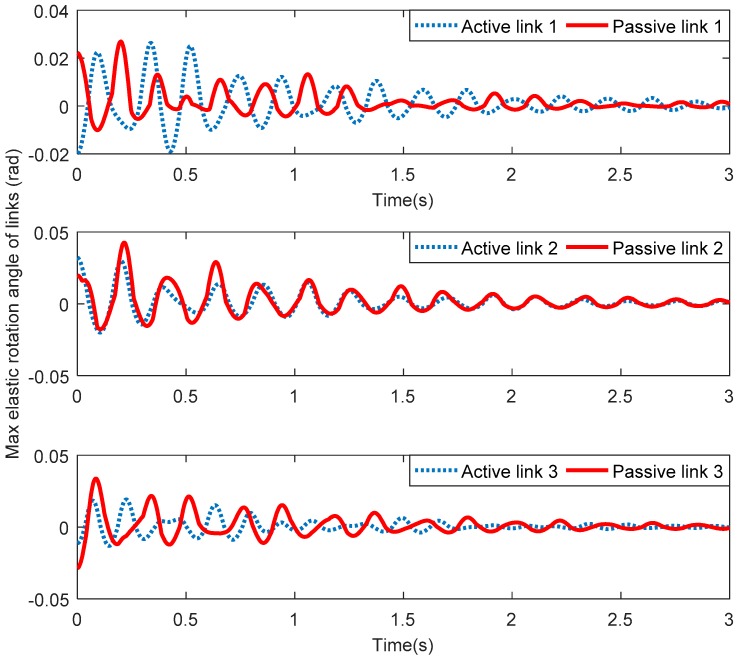
Maximum elastic rotation angles of active links and passive links.

**Figure 9 sensors-18-04145-f009:**
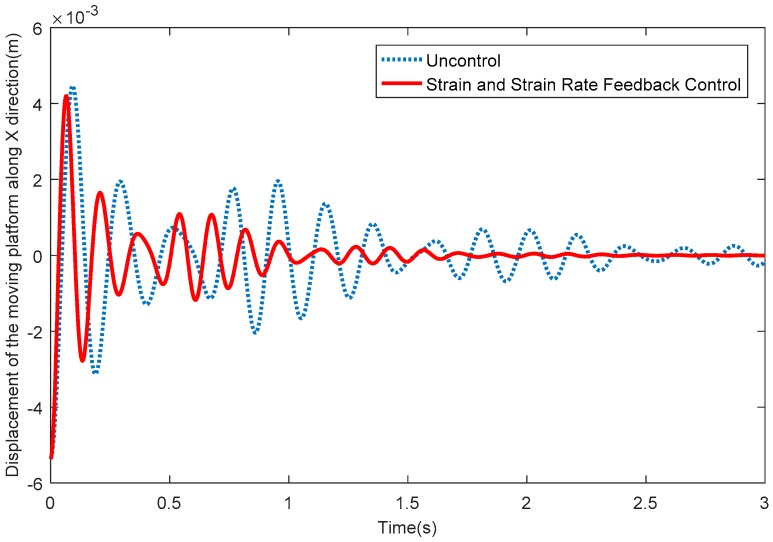
Elastic displacements of the moving platform along the *X* direction.

**Figure 10 sensors-18-04145-f010:**
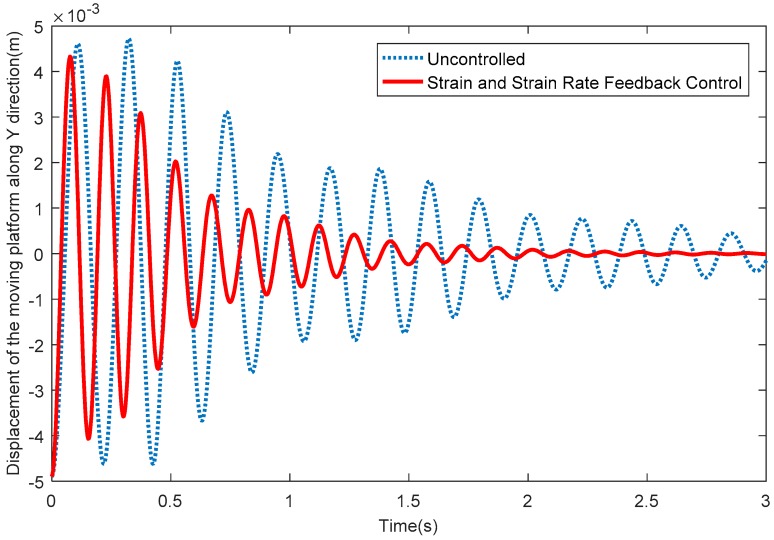
Elastic displacements of the moving platform along the *Y* direction.

**Figure 11 sensors-18-04145-f011:**
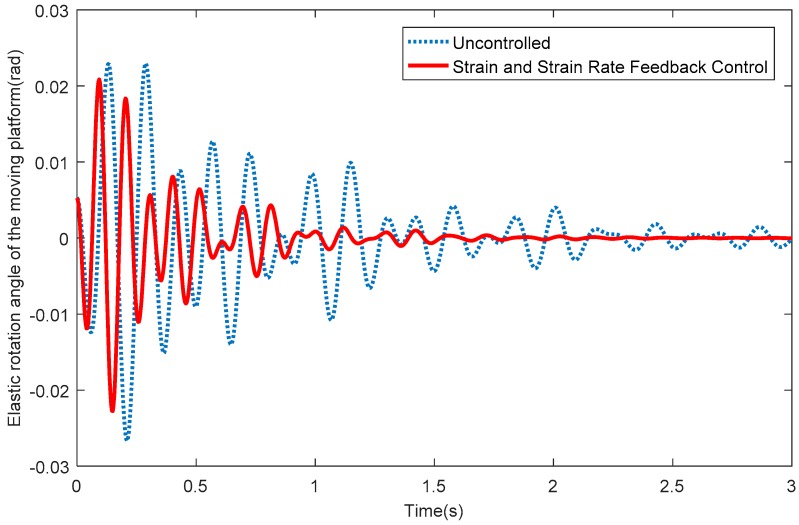
Elastic rotation angles of the moving platform in the XY plane.

**Figure 12 sensors-18-04145-f012:**
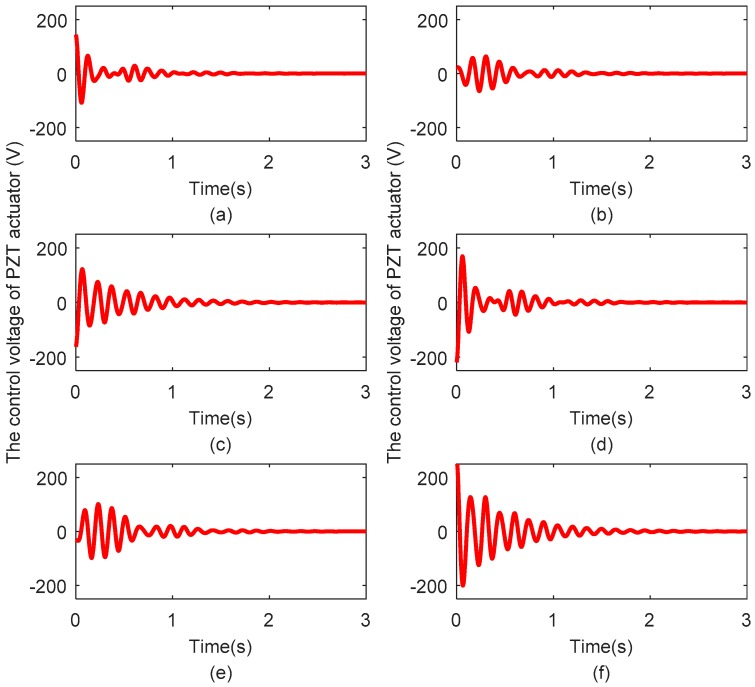
(**a**–**f**) Control voltages exerted on PZT actuator pairs 1, 2, 3, 4, 5, and 6, respectively.

**Figure 13 sensors-18-04145-f013:**
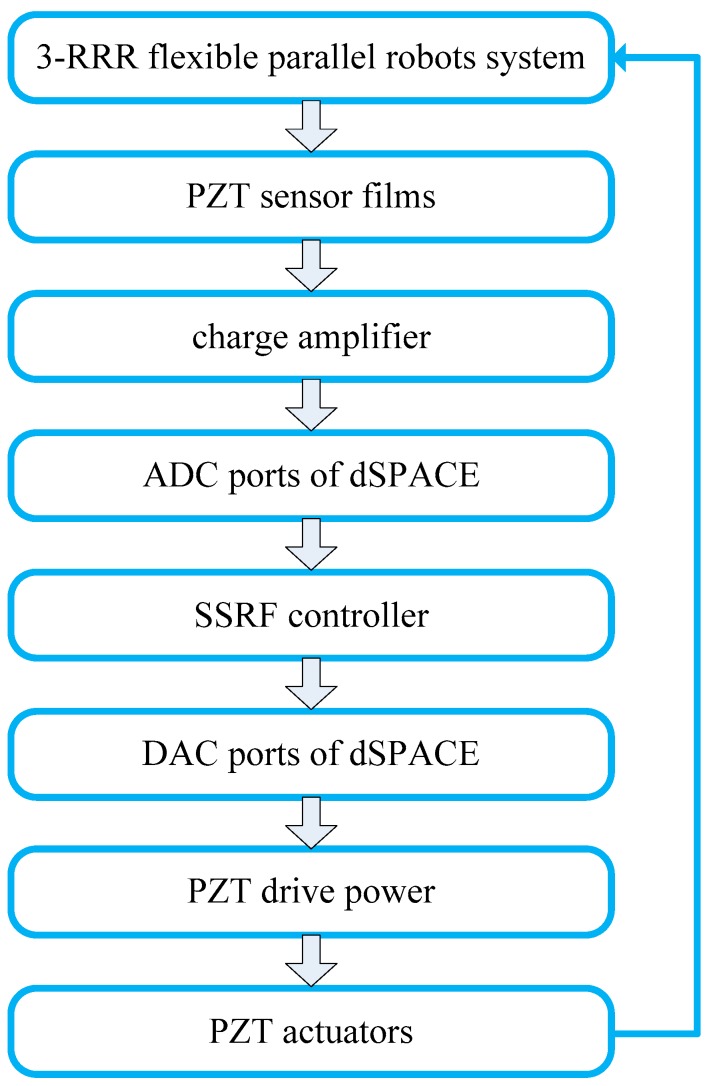
Experiment flow diagram.

**Figure 14 sensors-18-04145-f014:**
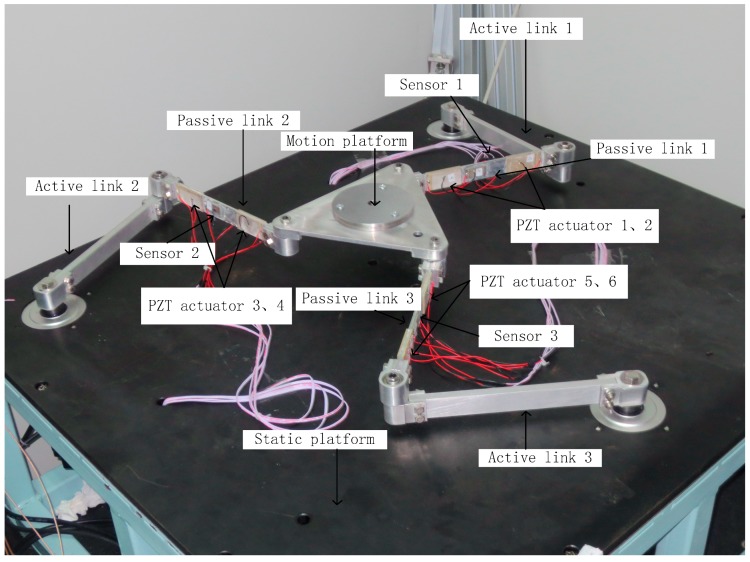
Experimental setup of the active vibration control system.

**Figure 15 sensors-18-04145-f015:**
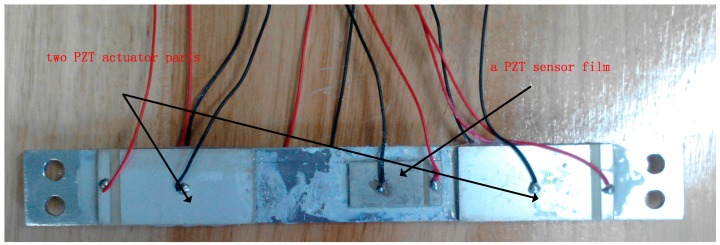
Smart beam.

**Figure 16 sensors-18-04145-f016:**
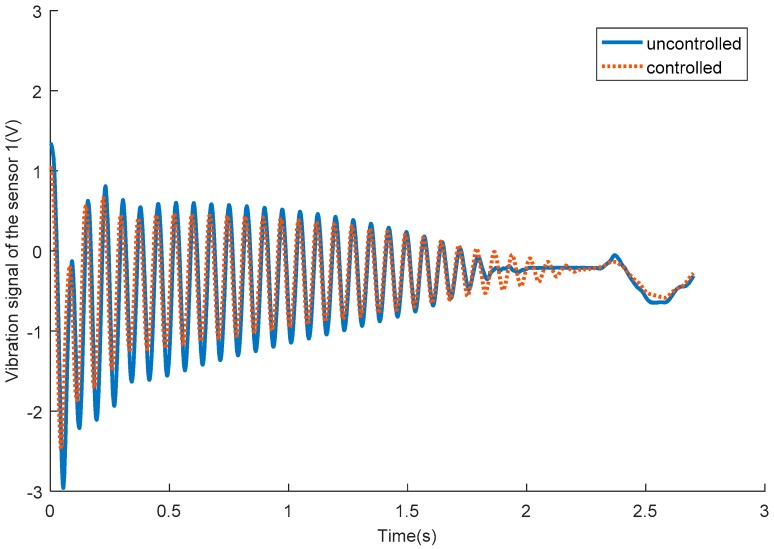
The vibration signals of passive flexible link 1 for the uncontrolled and controlled systems.

**Figure 17 sensors-18-04145-f017:**
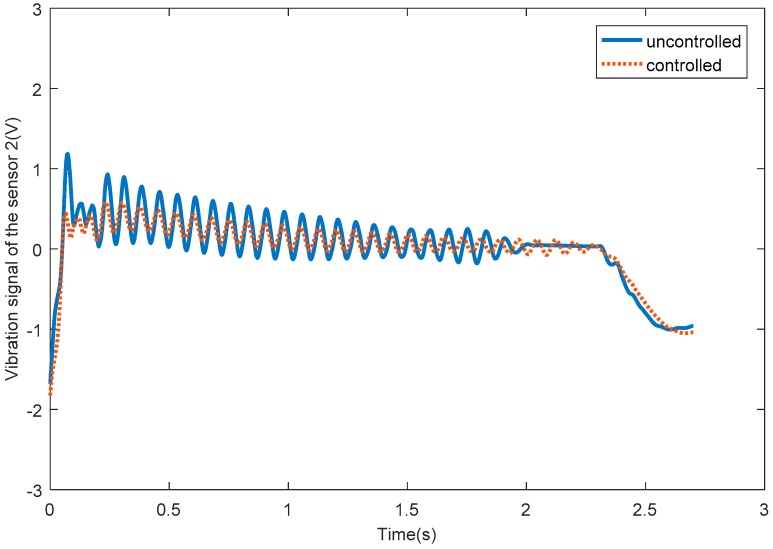
The vibration signals of passive flexible link 2 for the uncontrolled and controlled systems.

**Figure 18 sensors-18-04145-f018:**
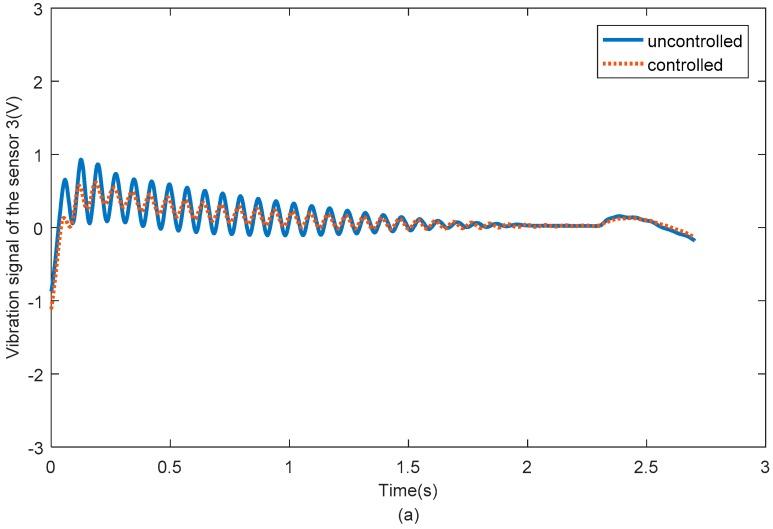
The vibration signals of passive flexible link 3 for the uncontrolled and controlled systems.

**Figure 19 sensors-18-04145-f019:**
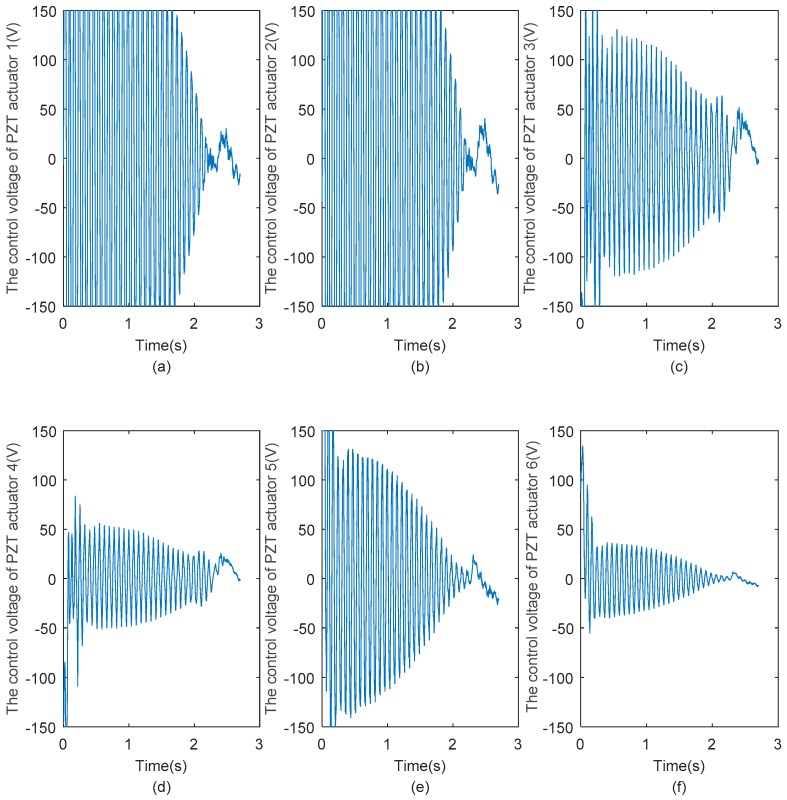
(**a**–**f**) Control voltages exerted pairs of PZT actuators 1, 2, 3, 4, 5, and 6, respectively.

**Table 1 sensors-18-04145-t001:** Parameters of the components of the 3-RRR system.

	Active links	Passive links	PZT actuator	PZT sensor
Length (mm)	0.254	0.252	0.05	0.03
Width (mm)	0.025	0.025	0.025	0.015
Thickness (mm)	0.01	0.003	0.002	0.001
Young’s modulus (MP)	0.7102 × 10^5^		1.17106 × 10^5^	
Density ρ (kg/m^3^)	2712		~	
Piezoelectric constant	~		1.86 × 10^−10^	
Poisson’s ratio	0.3			

**Table 2 sensors-18-04145-t002:** Data concerning the consumption of time (s).

	Displacements in the *X* Direction (m)		Displacements in the *Y* Direction (m)		Elastic Rotation Angles (rad)	
	<10^−3^	<10^−4^	<10^−3^	<10^−4^	<10^−2^	<10^−3^
Uncontrolled	1.713	>3	2.553	>3	1.173	>3
SSRF	0.7567	1.81	1.127	2.183	0.5233	1.44

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
