# Peer review of "Study on Residual Vibration Suppress of a 3-DOF Flexible Parallel Robot Mechanism"

_sensors, 2018, doi:10.3390/s18124145_

Reviewer 1 Report

This paper focused active vibration control of flexible structures which are links of the robot mechanism. This reviewer strongly recommend the authors to submit this paper to the journals related to vibration or/and control such as Journal of Vibration and Control. Other comments are summarized as follows.

There is no technical originality. The control method of the strain and strain rate feedback has been used by many researchers. Moreover, the use of PZT as an actuator and sensor is not impressive.

The robust stability of the system should be solidly proved.

The spillover problems due to the residual vibration modes need to be treated in details.

Control performance in simulation results is fairly good, but not impressive in experimental results. This should be clarified.

The authors cited their works too many. The authors should survey and cite references done by other researchers. Then, the mian difference from the previous works should be addressed in terms of control method or/and dynamic modeling.

Author Response

Dear Editors and reviewer:

Thank you for your letter and for the reviewers’ comments concerning our manuscript entitled “Study on Residual Vibration Suppress of a 3-DOF Flexible Parallel Robot Mechanism” (ID: sensors-385647). Those comments are all valuable and very helpful for revising and improving our paper, as well as the important guiding significance to our researches. We have studied comments carefully and have made correction which we hope meet with approval. Revised portion are marked in blue in the paper. The main corrections in the paper and the responds to the reviewer’s comments in an attached PDF document:

Reviewer 2 Report

The authors deal with the problem of reducing vibration in a 3-DoF planar flexible parallel robot. In order to accomplish their purpose, a strain and strain rate feedback PD controller is employed using PZT patches as sensors and actuators. They briefly introduce the adopted model, based on finite element method, and then study the effectiveness of the proposed control both with numerical simulations and experiments.

This is an interesting work dealing with a very significant subject, so the reviewer approves of its publication. The reviewer finds no fault whatsoever with the methods, theoretical and experimental analysis, or conclusions. The work is fundamentally sound.

Some observations might however help in further improving the quality of the submission.

The English must be revised because sometimes it is difficult to understand the meaning of the sentences. For example in the abstract, please rephrase: "Then, the experiment study are executed, one scheme that three passive flexible links are controlled at the same time is designed." 

The same for "active vibration control of flexible parallel robots mechanism is less studied due to its complicated." And so on...

Please explain the meaning of "KED" assumption in the introduction.

In the introduction, speaking of piezoelectric sensors, the authors state the good linearity of them. This is not completely true because those sensors exhibit a hysteric behaviour!

Some more information it is required for the trajectory tracking of the point P by means of the active rigid links. How do the motors impose the desired motion?

In the introduction, the reviewer suggests to mention papers related to the use of piezoelectric patches simultaneously as sensor and actuator for reducing vibrations in a wide range of frequencies (as done e.g. in [1,2,3])

[1] A. Preumont, J.-P. Dufour, and C. Malekian, Active damping by a local force feedback with piezoelectric actuators, Journal of Guidance, Control, and Dynamics, 15(2): 390–395, 1992.

[2] Giorgio I., Culla A. and Del Vescovo D. (2009) Multimode vibration control using several piezoelectric transducers shunted with a multiterminal network. Archive of Applied Mechanics, 79, 859–879.

[3] M.F. Lumentut, I.M. Howard. Effect of shunted piezoelectric control for tuning piezoelectric power harvesting system responses–analytical techniques. Smart Mater. Struct., 24(10), 105029, 2015.

Moreover, in the opinion of the reviewer, some more works on reduction of vibrations for a wide range of frequencies with other kinds of control different from the simple PD should be mentioned in the introduction (see e.g. [4]).

[4] Giorgio, I. & Del Vescovo, D. (2018) Non-Linear Lumped-Parameter Modeling of Planar Multi-Link Manipulators with Highly Flexible Arms. Robotics, 7(4) 13 pages. (DOI: 10.3390/robotics7040060)

Author Response

Dear Editors and reviewer:

Thank you for your letter and for the reviewers’ comments concerning our manuscript entitled “Study on Residual Vibration Suppress of a 3-DOF Flexible Parallel Robot Mechanism” (ID: sensors-385647). Those comments are all valuable and very helpful for revising and improving our paper, as well as the important guiding significance to our researches. We have studied comments carefully and have made correction which we hope meet with approval. Revised portion are marked in blue in the paper. The main corrections in the paper and the responds to the reviewer’s comments in an attached PDF document:

Round  2

Reviewer 1 Report

The authors have revised well based on the comments, but this reviewer is still sure that the scope of this paper is not suitable to Sensors. Even though the PZT is used as a sensor, there is no impressive results in the sense of sensors or sensed signals.

I am sure that this paper is well qualified for vibration control journal instead of the Sensors.